# EK100 and Antrodin C Improve Brain Amyloid Pathology in APP/PS1 Transgenic Mice by Promoting Microglial and Perivascular Clearance Pathways

**DOI:** 10.3390/ijms221910413

**Published:** 2021-09-27

**Authors:** Huey-Jen Tsay, Hui-Kang Liu, Yueh-Hsiung Kuo, Chuan-Sheng Chiu, Chih-Chiang Liang, Chen-Wei Chung, Chin-Chu Chen, Yen-Po Chen, Young-Ji Shiao

**Affiliations:** 1Institute of Neuroscience, School of Life Science, National Yang-Ming Chiao Tung University, Taipei 112, Taiwan; hjtsay@ym.edu.tw; 2National Research Institute of Chinese Medicine, Ministry of Health and Welfare, Taipei 112, Taiwan; hk.liu@nricm.edu.tw; 3Program in Clinical Drug Development of Chinese Medicine, Taipei Medical University, Taipei 112, Taiwan; 4Department of Chinese Pharmaceutical Sciences and Chinese Medicine Resources, China Medical University, Taichung 404, Taiwan; kuoyh@mail.cmu.edu.tw; 5Department of Biotechnology, Asia University, Taichung 413, Taiwan; 6Chinese Medicine Research Center, China Medical University, Taichung 404, Taiwan; 7Institute of Biopharmaceutical Science, National Yang-Ming Chiao Tung University, Taipei 112, Taiwan; vicvic1019@gmail.com; 8Institute of Anatomy and Cell Biology, National Yang-Ming Chiao Tung University, Taipei 112, Taiwan; tony0078013@yahoo.com.tw; 9Institute of Traditional Medicine, National Yang-Ming Chiao Tung University, Taipei 112, Taiwan; patrick831226@icloud.com; 10Biotech Research Institute, Grape King Bio Ltd., Taoyuan City 320, Taiwan; gkbioeng@grapeking.com.tw (C.-C.C.); yp.chen@grapeking.com.tw (Y.-P.C.)

**Keywords:** Alzheimer’s disease, APPswe/PS1dE9 transgenic mice, EK100, antrodin C, amyloid plaque, microglia, perivascular clearance

## Abstract

Alzheimer’s disease (AD) is characterized by the deposition of β-amyloid peptide (Aβ). There are currently no drugs that can successfully treat this disease. This study first explored the anti-inflammatory activity of seven components isolated from *Antrodia cinnamonmea* in BV2 cells and selected EK100 and antrodin C for in vivo research. APPswe/PS1dE9 mice were treated with EK100 and antrodin C for one month to evaluate the effect of these reagents on AD-like pathology by nesting behavior, immunohistochemistry, and immunoblotting. Ergosterol and ibuprofen were used as control. EK100 and antrodin C improved the nesting behavior of mice, reduced the number and burden of amyloid plaques, reduced the activation of glial cells, and promoted the perivascular deposition of Aβ in the brain of mice. EK100 and antrodin C are significantly different in activating astrocytes, regulating microglia morphology, and promoting plaque-associated microglia to express oxidative enzymes. In contrast, the effects of ibuprofen and ergosterol are relatively small. In addition, EK100 significantly improved hippocampal neurogenesis in APPswe/PS1dE9 mice. Our data indicate that EK100 and antrodin C reduce the pathology of AD by reducing amyloid deposits and promoting nesting behavior in APPswe/PS1dE9 mice through microglia and perivascular clearance, indicating that EK100 and antrodin C have the potential to be used in AD treatment.

## 1. Introduction

Alzheimer’s disease (AD) is characterized by a progressive decline in cognitive abilities that can ultimately affect daily living activities. Extracellular deposition of β-amyloid (Aβ) plaques and intracellular neurofibrillary tangles are considered the main pathological hallmark. The imbalance between Aβ production and clearance causes Aβ to accumulate in the central nervous system (CNS), which in turn induces Aβ plaque formation, increases neuroinflammation, and alters adult hippocampal neurogenesis [1].

There is increasing evidence that the clearance of Aβ from the brain in AD is impaired. Therefore, there is an increasing need to study the Aβ clearance mechanism as a potential target for AD treatment. As we all know, Aβ can be cleared from the brain in many ways, such as being degraded by proteases, transported through the blood–brain barrier (BBB), and cleared by the flow of interstitial fluid (ISF) and cerebrospinal fluid (CSF)) [2]. According to recent studies, two types of perivascular scavenging systems are related to the movement of ISF and CSF, namely intramural peri-arterial drainage (IPAD) and perivascular CSF inflow [2,3]. In IPAD, the parenchymal ISF containing Aβ exits the brain and enters the cervical lymph nodes along the basement membrane (BM) of capillaries and arterial smooth muscle cells [3]. IPAD failure can cause cerebral amyloid angiopathy (CAA), where Aβ mainly accumulates in capillaries and arterial smooth muscle cells BM [4]. The development of AD animal models is crucial for elucidating the mechanisms and the etiology of the disease in order to identify efficient therapies. Therefore, animal models overexpressing Aβ or tau markers were extensively generated [5]. Animal models were first used to understand AD mechanisms, but above all, they were used to test drugs preventing cognitive deficits for therapeutic interventions. However, none of the current developed animal models is able to entirely reproduce the AD pathology, since the deposition of Aβ and its consequence are not limited to the central nervous system [6,7,8].

Neuroinflammation can damage neurons and promote Aβ aggregation, so anti-inflammatory has become the target of AD treatment [9,10]. However, studies have shown that the early activation of M2-like microglia in AD exhibits neuroprotective functions by promoting Aβ phagocytosis and clearance, but Aβ-activated M1-like microglia down-regulate Aβ clearance, thereby promoting Aβ aggregation and neurodegeneration [11]. It has been proposed that pathologically induced COX-2 activity can induce memory deficits in APPswe/PS1dE9 (APP/PS1) transgenic mice [12]. Therefore, the potential beneficial effects of non-steroidal anti-inflammatory drugs (NSAIDs) such as ibuprofen on the treatment of AD have also attracted people’s attention [13].

*Antrodia cinnamomea* (Syn. *Antrodia camphorata*) is a mushroom unique to Taiwan [14] that has been safely used in humans in clinical trials (ClinicalTrials.gov (accessed on 1 September 2021) Identifier:NCT01007656). For a long time, the fruit body of *A. cinnamomea* has been widely used as a traditional medicine for the treatment of many medical diseases [15]. At present, the cultured mycelium of *A. cinnamomea* and its anti-inflammatory components, such as ergosta-7,9(11), 22-trien-3β-ol (also known as EK100), antrodin C, and antroquinonol can also be used for medicinal purposes. Antroquinonol activates the nuclear factor erythroid-2 related factor-2 (Nrf2)-dependent cellular antioxidant defense system, which has antioxidant and anti-inflammatory effects in peripheral diseases and AD animal models [16]. 

In this study, six phytosteroids (i.e., EK100, anticin K, eburicoic acid, dehydroeburicoic acid, sulfurenic acid, and dehydrosulfurenic acid) (Figure 1A) and a maleimide derivative (i.e., antrodin C) (Figure 1B) were isolated from the mycelium or fruit body of *Antrodia cinnamomea* for anti-inflammatory research. In order to be used as structural control, ergosterol and ibuprofen were also included in this study (Figure 1A,B).

In previous studies, it was found that EK100 has an anti-inflammatory effect on the liver of chronic alcohol-fed mice [17] and improves brain damage in ischemic stroke through neurogenesis [18]. Ergosterol is an isomer of EK100 (Figure 1B), which has an anti-inflammatory effect on the nitric oxide inhibitory activity in macrophages [19]. Antrodin C (3-isobutyl-4-[4-(3-methyl-2-butenyloxy)phenyl]-1H-pyrrol-1-ol-2,5-dione) has anti-inflammatory activity and can inhibit macrophage nitric oxide production [20]. By activating the Nrf2-dependent cellular antioxidant defense system, it can effectively intervene in diabetes-related cardiovascular diseases [21]. Antcin K, but not EK100, activates peroxisome proliferator-activated receptor (PPAR)α in cell-based transactivation studies [22]. Eburicoic acid and dehydroeburicoic acid-treated mice reduced hyperglycemia, hypertriglyceridemia, hyperinsulinemia, hyperleptinemia, and hypercholesterolemia induced by a high-fat diet [23]. Sulfuric acid showed protective effects on type 1 diabetes and hyperlipidemia in diabetic mice induced by streptozotocin [24]. However, the anti-Alzheimer’s disease effect of these compounds has never been studied.

APP/PS1 transgenic mice co-expressing Swedish mutant human APP695 and mutant human presenilin 1 (PS1) (in which exon 9 is deleted) [25] exhibit pathological and behavioral changes similar to AD, including amyloid in the brain accumulation of plaques, degeneration of the cholinergic system, and impaired exploratory behavior and spatial memory [26]. As early as 3 to 5 months of age, APP/PS1 mice have increased Aβ production and plaque formation [27], and impairments in spatial learning and memory are also observed at 6 months of age [28,29]. In order to verify the effects of the four compounds on behavioral disorders in APP/PS1 mice, we focused on species-specific nesting activities, because it is multi-brain-dependent spontaneous [30], which has been considered similar to activities of daily living (ADL) skills [31]. Clinically, ADL disorder is a pathological manifestation of AD [32], and this pathological manifestation also appears in APP/PS1 mice [29]. In addition, it was found that the hippocampal neurogenesis of APP/PS1 mice was damaged at 3 to 6 months of age [33]. Our previous studies have shown that anti-inflammatory effects can promote hippocampal neurogenesis [29], so we hypothesized that the decline in microglia activation may subsequently promote hippocampal neurogenesis.

The purpose of this study is to study the effects of EK100, antrodin C, ergosterol, and ibuprofen on AD-related pathology in APP/PS1 transgenic mice.

## 2. Results

### 2.1. EK100 and Antrodin C Confer Anti-Inflammatory Effects on BV2 Microglia

The structure of EK100 and antrodin C as well as their related compounds used in this study are shown in Figure 1. 

In order to determine the anti-inflammatory effects of *A. cinnamomea* mycelium components, BV2 cells, a cell line derived from primary mouse microglia cells, were incubated with various concentrations of *A. cinnamomea* mycelium components including ethanol extract (CA-Et) and seven isolated compounds for 30 min, and then, the cells were activated with LPS. Ergosterol (an isomer of EK100) and ibuprofen (a non-steroidal anti-inflammatory drug) were used as controls. The production of nitric oxide in the conditioned medium and the reduction of MTT in the cell were measured after 24 h incubation (Table 1). CA-Et with a non-cytotoxic concentration of 50 μg/mL significantly inhibited the production of nitric oxide in the vehicle-treated control group by 40% (F(5,12) = 59.70; *p* < 0.0001, *n* = 4). At subtoxic concentrations, antitoxin K, ethylene propionic acid/dehydrobutyric acid mixture, sulfuric acid/dehydrosulfuric acid mixture, and ibuprofen have no anti-inflammatory effects. Compared with the vehicle control, EK100 (20 μM), antrodin C (100 μM), and ergosterol (10 μM) reduced nitric oxide production by 34.0% (F(3, 16) = 44.97; *p* < 0.0001, *n* = 4), 40.0% (F(3, 16) = 198.67; *p* < 0.0001, *n* = 4), and 40.2% (F(3, 12) = 48.87; *p* < 0.0001, *n* = 4), respectively.

### 2.2. EK100 and Antrodin C Improve Nesting Behavior of APP/PS1 Mice 

We chose APP/PS1 transgenic mice (an AD animal model) to check that the effect of two anti-inflammatory components of *A. cinnamomea* (i.e., EK100 and antrodin C) and two control compounds (i.e., ergosterol and ibuprofen) on AD pathology. APP/PS1 mice (five-month-old, male and female) were orally administered EK100, antrodin C, ergosterol, and ibuprofen (30 mg⋅kg^−1^⋅day^−1^) or vehicle for 30 days, and body weight were measured weekly. The treatments did not significantly change body weight, indicating that these treatments have no obvious side effect. Nesting behavior involves a wide network of brain regions and has previously been used to assess the activities of daily living (ADL) skills in AD transgenic mice [29]. Compared with wild-type mice, APP/PS1 mice administered vehicle showed defects in nesting behavior, as assessed by nest score (1.33 ± 0.01 vs. 4.88 ± 0.09, *p* < 0.001, *n* = 6) and unthreaded Nestlet (3.77 ± 0.31 vs. 0.87 ± 0.32, *p* < 0.0001, *n* = 6). The use of EK100, antrodin C, and ibuprofen can significantly restore impaired nesting behavior (3.85 ± 0.57 vs. 1.33 ± 0.01, *p* < 0.01, *n* = 6, 3.70 ± 0.58 vs. 1.33 ± 0.01, *p* < 0.01, *n* = 6, 2.50 ± 0.58 vs. 1.33 ± 0.01, *p* < 0.05, *n* = 6, respectively) (Figure 2). However, ergosterol has no effect on nesting behavior.

### 2.3. EK100 and Antrodin C Reduce the Number and Burden of Amyloid Plaques in the Brains of APP/PS1 Mice

It is known that plaques can be clearly observed in the brains of APP/PS1 mice at 6 months of age [29]. Therefore, the effect of the four compounds on the number and size distribution of plaque were detected by immunostaining with AB10 antibody. The plaque number and burden were calculated using MetaMorph software. The plaque number and burden were significantly reduced after 30-day administration of EK100 or antrodin C. Compared with vehicle-treated mice, EK100 and antrodin C decreased plaque number by 45.4% (170.3 ± 19.04 vs. 272.6 ± 30.89, *p* < 0.05, *n* = 5) and 43.7% (181.4 ± 19.76 vs. 272.6 ± 30.89, *p* < 0.05, *n* = 5), respectively (Figure 3A,B; Appendix A), and they decreased plaque burden by 37.5% (1.31 ± 0.24 vs. 2.13 ± 0.19, *p* < 0.05, *n* = 5) and 53.9% (0.98 ± 0.12 vs. 2.13 ± 0.19, *p* < 0.01, *n* = 5), respectively (Figure 3A,C). In contrast, ergosterol and ibuprofen did not have any significant effect on the number of plaques, and ibuprofen even increased the plaque load by 31.6% (2.80 ± 0.06 vs. 2.13 ± 0.19, *p* < 0.05, *n* = 5).

Since the decrease in the number of plaques may be due to a decrease in Aβ levels in the brain, we subsequently measured Aβ levels in the hippocampus. However, after all four treatments, there was no significant change in Aβ levels in the hippocampus (Appendix A). In contrast, serum Aβ1-42 levels significantly decreased after the ergosterol and ibuprofen treatment, while serum levels of Aβ1-40 significantly increased after antrodin C treatment (Appendix A).

### 2.4. EK100 and Antrodin C Promote Aβ Perivascular Deposition in the Brain of APP/PS1 Mice 

Since Aβ in the ISF of the brain can be removed from the brain through the glymphatic perivenous drainage pathway and/or IPAD pathway [34], perivascular Aβ deposition is detected in both the cortex and hippocampus. We found that the distribution of Aβ deposits changed from amyloid plaques to Aβ deposits around blood vessels after treatment with EK100 and antrodin C (Figure 4). The calculation of Aβ deposition in the perivascular area is expressed as a percentage of the total deposition (Figure 4B). The results showed that EK100 and antrodin C increased the perivascular Aβ deposition in the cortex by 22.86% (63.97 ± 5.38 vs. 41.12 ± 3.71, *p* < 0.01, *n* = 5 vs. 6) and 19.89% (61.00 ± 0.84 vs. 41.12 ± 3.713, *p* < 0.01, *n* = 5 vs. 6), respectively. These changes were found in both capillaries and arteries (Appendix A). Similarly, EK100 and antrodin C increase the perivascular Aβ deposition in the hippocampus by 31.67% (76.48 ± 13.05 vs. 44.80 ± 6.52, *p* < 0.05, *n* = 5 vs. 6) and 28.21% (73.01 ± 4.84 vs. 44.80 ± 6.52, *p* < 0.05, *n* = 5 vs. 6), respectively.

### 2.5. EK100 and Antrodin C Reduce the Number of Glial Cluster and Glial Activation in the Brain of APP/PS1 Mice

M2-microglia phagocytosis is another Aβ clearance pathway. Therefore, changes in the phenotype of microglia may also help reduce amyloid plaques. After 30 days of treatment with EK100 and antrodin C, the number of clusters formed by PAM decrease significantly (Figure 3A,C). Compared with vehicle-treated mice, EK100 treatment reduced the number of clusters formed by PAM by 35.5 ± 9.6% (37.40 ± 2.50 vs. 57.83 ± 2.07, *p* < 0.001, *n* = 5). The less effective is that antrodin C treatment reduces the number of clusters containing PAM by 34.2 ± 18.5% (41.40 ± 5.66 vs. 57.83 ± 2.07, *p* < 0.05, *n* = 5). In contrast, ergosterol and ibuprofen had no significant effect on the number of clusters formed by PAM.

On the other hand, compared with vehicle-treated mice, EK100 treatment reduced number of clusters formed by PAA by 42.0 ± 7.9% (34.80 ± 2.11 vs. 59.67 ± 1.84, *p* < 0.0001, *n* = 5) (Figure 3A,D). The less effective is that antrodin C treatment reduces the number of clusters containing PAA by 34.4 ± 20.1% (41.20 ± 4.62 vs. 59.67 ± 1.84, *p* < 0.01, *n* = 5). In contrast, ergosterol and ibuprofen had no significant effect on the number of clusters formed by PAA and PAM.

In order to determine the changes in microglia associated with activated plaques after drug treatment, a scatter plot of the fluorescence intensity of small and medium plaques (<21 pixels) and Iba-1 was drawn (Appendix A). Linear regression analysis showed that the slope of the regression equation of EK100, antrodin C, and ibuprofen groups were significantly different from that of the vehicle group. Linear regression for the vehicle group is y = 0.51x + 0.76, R^2^ = 0.06; for the EK100 group is y = 1.12x + 0.47, R^2^ = 0.23 (*p* < 0.01 for different in slope); for the antrodin C group is y = 1.05x + 3.32, R^2^ = 0.18 (*p* < 0.01 for different in slope); for the ergosterol group is y = 0.84x + 4.47, R^2^ = 0.21 (*p* = 0.13 for different in slope); for the ibuprofen group is y = 0.89x + 0.95, R^2^ = 0.21 (*p* < 0.01 for different in slope). This result indicates that EK-100, antrodin C, and ibuprofen, but not ergosterol, changed the relationship between plague and microglia activation.

In order to determine glial activation associated with plaques after drug treatment, the representative fluorescent images (*n* = 34–56) are captured (Figure 3E). The immunointensity (IR) ratio of Iba-1 to AB10 (Figure 3F) and GFAP to AB10 (Figure 3G) of the captured images are calculated by ImageJ software. The results showed that EK100 and antrodin C increase the IR ratio of Iba-1 to AB10 by 45.12% (2.63 ± 0.36 vs. 1.82 ± 0.20, *p* < 0.05, *n* = 46 vs. 54) and 59.45% (2.89 ± 0.39 vs. 1.82 ± 0.20, *p* < 0.01, *n* = 30 vs. 54). On the other hand, EK100 and antrodin C did not change the IR ratio of GFAP to AB10. However, ergosterol decrease the IR ratio of GFAP to AB10 by 35% (2.35 ± 0.23 vs. 3.55 ± 0.29, *p* < 0.05, *n* = 53).

### 2.6. EK100 and Antrodin C Reduce Non-Clustered Activation of Glia in the Hippocampus of APP/PS1 Mice

Previously, we found that compared with wild-type mice, APP/PS1 mice had a higher degree of astrocyte reactivity and microglia activation in areas unrelated to plaque [29]. Therefore, the immunoreactivity of the non-clustered glial cells in the hippocampus of each group was compared (Figure 5). The results show that compared with wild-type mice, APP/PS1 mice have a higher degree of glial non-cluster activation. Compared with vehicle-treated mice, EK100 and ibuprofen treatment reduced the IR of non-clustered astrocytes (NCA) in the hippocampal CA1 by 46.7% (9.49 ± 0.42 vs. 17.80 ± 1.98, *p* < 0.01, *n* = 5 vs. 6) and 47.8% (9.30 ± 0.50 vs. 17.80 ± 1.98, *p* < 0.05, *n* = 5 vs. 6), respectively (Figure 5A,B). In contrast, antrodin C and ergosterol did not show a significant effect on the IR of NCA in the hippocampus. On the other hand, EK100, antrodin C and ibuprofen treatment reduced the IR of non-clustered microglia (NCM) in hippocampal CA1 by 50.7% (5.87 ± 0.61 vs. 11.90 ± 1.03, *p* < 0.001, *n* = 5 vs. 6), 54.7 ± 9.7% (5.14 ± 0.67 vs. 11.90 ± 1.03, *p* < 0.001, *n* = 5 vs. 6) and 38.1% (7.37 ± 1.37 vs. 11.90 ± 1.03, *p* < 0.05, *n* = 5 vs. 6), respectively (Figure 5B). In contrast, ergosterol did not show a significant effect on the IR of NCM in the hippocampus. The similar changes were found in CA3 and DG areas.

Microglia have different morphological responses to changes in brain physiology, from hyper-ramified form to amoeba form [35]. Since microglia fine-tune the function of neurons and glial through cell-to-cell crosstalk [36], the morphology of microglia can be used as an indicator of a variety of cell functions and dysfunctions in the brain. The skeletal analysis was used for regional analysis of multiple microglia in the region of interest. The results shown that EK100, ergosterol, and ibuprofen increase the number of branches, junctions, and endpoints of microglia (Figure 5C,D). EK100, ergosterol, and ibuprofen treatment increased the branch number of microglia by 11.6% (5.56 ± 0.06 vs. 4.98 ± 0.09, *p* < 0.0001, *n* = 20), 18.3% (5.89 ± 0.12 ± 0.06 vs. 4.98 ± 0.09, *p* < 0.0001, *n* = 25 vs. 20), and 9.4% (5.45 ± 0.11 vs. 4.98 ± 0.09, *p* < 0.0001, *n* = 25 vs. 20), respectively. The similar changes were found in the intersection number and endpoint number of microglia.

Next, the combined immunostaining of Iba-1 and AB10 was used to check the microglia phagocytosis of Aβ. The results showed that antrodin C, ergosterol, and ibuprofen, significantly increased the accumulation of Aβ in microglia by 37.4% (32.18 ± 1.85 vs. 23.42 ± 2.41, *p* < 0.01, *n* = 32 vs. 25), 57.0% (36.76 ± 2.45 vs. 23.42 ± 2.41, *p* < 0.001, *n* = 25), and 40.1% (31.82 ± 2.60 vs. 23.42 ± 2.41, *p* < 0.05, *n* = 25) (Figure 5E,F). On the contrary, EK100 did not alter the accumulation of Aβ in microglia.

### 2.7. EK100 and Antrodin C Eliminated Nrf2 Overexpression in the Brain of APP/PS1 Mice

Nrf2 regulates the expression of phase II detoxification enzymes (including HO-1 and NQO-1) as well as antioxidant genes that protect cells from various damages through anti-inflammatory effects, thereby affecting the progression of the disease [37]. Therefore, the expression of HO-1 was detected. Compared with wild-type mice, APP/PS1 mice have increased HO-1 expression (Figure 6A). However, antrodin C and ergosterol significantly promoted the increase of HO-1 expression by 24.3% (0.92 ± 0.04 vs. 0.74 ± 0.03, *p* < 0.001, *n* = 57 vs. 66) and 24.3% (0.92 ± 0.04 vs. 0.74 ± 0.03, *p* < 0.001, *n* = 64 vs. 66), respectively (Figure 6B,C). In contrast, EK100 and ibuprofen significantly reduced the expression of HO-1 compared to vehicle therapy by 21.7% (0.58 ± 0.03 vs. 0.74 ± 0.03, *p* < 0.001, *n* = 71 vs. 66) and 12.2% (0.65 ± 0.03 vs. 0.74 ± 0.03, *p* < 0.01, *n* = 62 vs. 66), respectively. Both HO-1 and NQO-1 are mainly expressed in microglia and overexpressed in the PAM of APP/PS1 mice (Figure 6A). Antrodin C, ergosterol, and ibuprofen significantly promote the increase of NQO-1 expression by 50% (0.93 ± 0.04 vs. 0.62 ± 0.03, *p* < 0.0001, *n* = 45 vs. 57), 116% (1.34 ± 0.06 vs. 0.62 ± 0.03, *p* < 0.0001, *n* = 64 vs. 57), and 53.2% (0.95 ± 0.03 vs. 0.62 ± 0.03, *p* < 0.0001, *n* = 53 vs. 57), respectively (Figure 6B,C). In contrast, EK100 had no effect on NQO-1 expression compared with vehicle treatment.

Next, we examined the expression of Nrf2 in 6-month-old APP/PS1 mice. We found that APP/PS1 mice have increased Nrf2 expression compared to wild-type mice (Figure 6D,E), and treatment of mice with EK100, antrodin C, and ibuprofen significantly reduced Nrf2 expression by 63.9% (143.4 ± 30.43 vs. 396.8 ± 34.80, *p* < 0.001, *n* = 5 vs. 6), 73.0% (107.2 ± 11.39 vs. 396.8 ± 34.80, *p* < 0.001, *n* = 5 vs. 6), and 55% (178.5 ± 14.53 vs. 396.8 ± 34.80, *p* < 0.01, *n* = 5 vs. 6), respectively. Overexpressed Nrf2 is distributed throughout the brain, but it is not specifically expressed in microglia. We found that compared with wild-type mice, the expression of Nrf2 in neurons of APP/PS1 mice was significantly enhanced by 178.6% (143,362 ± 26,766 vs. 51,447 ± 7022, *p* < 0.01, *n* = 14 vs. 10) (Figure 6F,G). After the administration of EK100 and antrodin C, the overexpression of Nrf2 in neurons were decreased by 54.2% (65,729 ± 7769 vs. 143,362 ± 26,766, *p* < 0.05, *n* = 10 vs. 14) and 48.2% (74,224 ± 9136 vs. 143,362 ± 26,766, *p* < 0.05, *n* = 10 vs. 1), respectively. In contrast, EK100 and antrodin C did not affect the Aβ content in neurons.

### 2.8. EK100 Promote Hippocampal Neurogenesis and Dendritic Complexity in the Brain of APP/PS1 Mice

Compared with wild-type mice, the number of BrdU-positive proliferating type 2 progenitor cells and DCX-positive neonatal granule neurons in the subgranular zone (SGZ) of APP/PS1 mice were decreased by 51.3% (6.02 ± 0.74 vs. 12.36 ± 1.08, *p* < 0.001, *n* = 9 vs. 8) and 34.1% (20.76 ± 1.11 vs. 31.48 ± 2.03, *p* < 0.001, *n* = 9), respectively, and EK100 restored these decreases by 67.8% (10.10 ± 0.51 vs. 6.02 ± 0.74, *p* < 0.001, *n* = 10 vs. 9) and 38.9% (28.83 ± 2.41 vs. 20.76 ± 1.11, *p* < 0.001, *n* = 8 vs. 10) (Figure 7A–D). The number of BrdU- and DCX-double positive neurons was also decreased by 73.8% (3.20 ± 0.48 vs. 8.83 ± 1.01, *p* < 0.001, *n* = 8 vs. 9), and EK100 restored these decreases by (6.54 ± 0.55 vs. 3.20 ± 0.48, *p* < 0.001, *n* = 10 vs. 8).

Since dendritic growth is important for neuron integration in neurogenesis, we further analyzed the dendritic complexity of DCX-positive cells through laminar flow quantitative methods [38]. We found that the vehicle-treated transgenic mice had lower levels of secondary dendritic branches compared to wild-type mice (1.34 ± 0.05 vs. 1.66 ± 0.10, *p* < 0.01, *n* = 10 vs. 9). EK100 significantly increased the dendritic complexity of secondary dendritic branches by 40.3% (1.88 ± 0.14 vs. 1.34 ± 0.05, *p* < 0.01, *n* = 9 vs. 10) (Figure 7E,F).

## 3. Discussion

This study shows that two anti-inflammatory compounds, EK100 and antrodin C, selected from components isolated from *A. cinnamomea* mycelium, can reduce AD-like pathological changes in APP/PS1 mice in different ways when administered orally. BV2 cells activated by LPS were used to determine the anti-inflammatory effects of components isolated from *A. cinnamomea* mycelium (including CA-Et, anticin K, eburicoic acid, dehydroeburicoic acid, sulfurenic acid, dehydrosulfurenic acid, EK100, antrodin C, and ergosterol), and ibuprofen (an NSAID control). The results showed that the production of nitric oxide in BV2 cells activated by LPS was significantly inhibited by CA-Et, EK100, antrodin C, and ergosterol, but it was not affected by anticin K, eburicoic acid/dehydroeburicoic acid mixture, sulfurenic acid/dehydrosulfurenic acid mixture, and ibuprofen. Therefore, we examined the effects of EK100, antrodin C, ergosterol, and ibuprofen (an NSAID control) on the pathology of APP/PS1 transgenic mice, including changes in nesting behavior, plaque deposition, perivascular deposition, nerves glial activation, and the Nrf2/HO-1/NQO-1 signaling pathway. Moreover, the effect of EK100 on neurogenesis was also examined. 

The results of the animal study are summarized in Table 2. In the behavior test, EK100, antrodin C, and ibuprofen, but not ergosterol, significantly improve the nesting ability of APP/PS1 mice. Among the effects related to Aβ deposition, EK100 and antrodin C can effectively reduce the number and load of plaques, promote Aβ deposition around blood vessels, and reduce the accumulation of PAA and PAM. Among the changes in PAM activation, EK100 significantly increased the expression of Iba-1 but did not change the ratios of HO-1/Iba-1 and NQO-1/Iba-1; ADC significantly increased Iba-1 and the HO-1/Iba-1 and NQO-1/Iba-1 ratios; ergosterol significantly increased the ratios of HO-1/Iba-1 and NQO-1/Iba-1 but did not increase the expression of Iba-1; and ibuprofen significantly increased the NQO-1/Iba-1 ratio and reduced the NQO-1/Iba-1 ratio without increasing the expression of Iba-1. There are no compounds that can modulate the expression of GFAP in PAA. In the activation of non-plaque-associated glial cells, EK100, antrodin C, and ibuprofen significantly increased the expression of Iba-1 in NCM, while antrodin C, ergosterol, and ibuprofen significantly increased the intracellular accumulation of Aβ in NCM. On the other hand, EK100, ergosterol, and ibuprofen significantly increased the ramification of NCM, while EK100 and ibuprofen can significantly reduce the expression of GFAP in NCA. In the response of non-plaque-related neurons, EK100, antrodin C, and ibuprofen significantly reduced the expression of Nrf2 in cortical neurons. Finally, EK100 has also been shown to significantly promote hippocampal neurogenesis. 

In order to verify the effects of the four compounds on behavioral disorders in APP/PS1 mice, we focused on species-specific nesting activities, because it is multi-brain-dependent spontaneous [30], which has been considered similar to ADL skills [31]. Clinically, ADL disorder is a pathological manifestation of AD [32], and this pathological manifestation also appears in APP/PS1 mice [29]. In this study, we found that EK100, antrodin C, and ibuprofen can alleviate the defects of nesting behavior. These results indicate that the administration of EK100, antrodin C, and ibuprofen may have the potential to restore multiple brain injury in APP/PS1 mice. 

Then, we studied the effects of four compounds on amyloid plaque deposition and Aβ perivascular deposition. EK100 and antrodin C reduced the number of plaques but did not reduce the levels of Aβ in the hippocampus and serum, indicating that the reduction in the number of plaques was not due to the inhibition of Aβ production. The use of SH-SY5Y-APP695 cells also confirmed the ineffective inhibition of Aβ accumulation by EK100 and antrodin C. Therefore, the effect of EK100 and antrodin C on reducing the number of plaques can be attributed to the removal of Aβ rather than the formation of Aβ.

Recent evidence suggests that impaired clearance may be the driving force behind sporadic AD [39]. Microglia may promote Aβ clearance through phagocytosis [40]. Although it is obvious that astrocytes and microglia accumulate around amyloid plaques in AD, it is still elusive whether they are mainly attracted by amyloid deposits or only by the plaque-related damaged neurite response [41]. Microglia activation is highly correlated with the accumulation of Aβ, because activated microglia are found to surround the plaque. Therefore, Aβ can be cleared by phagocytosis or proteolytic degradation [42]. Studies have shown that limiting the accumulation and phagocytosis of microglia will increase Aβ deposition, thus highlighting the functional impact of phagocytosis [43].

It is worth noting that EK100, antrodin C, and ibuprofen can effectively improve nesting behavior and inhibiting NCM activation, indicating that these two events are related in APP/PS1 mice. Microglia exhibit a variety of phenotypic states, from pro-inflammatory M1 phenotype to the alternative activation M2 phenotype, especially under chronic inflammatory conditions [44]. In vitro evidence suggests that the phagocytic ability of microglia is inhibited in AD [45]. The activity of M1-like reactive microglia induced by LPS in Aβ phagocytosis is significantly reduced, and this reduction can be rescued by IL-4 induced activation of M2-like microglia [45]. This phenomenon leads to the hypothesis that the accumulation of Aβ in AD may be due to changes in the phenotype of microglia. Therefore, the regulation of M2-like reactive microglia by microglia may have potential benefits in the treatment of AD.

Nrf2 is overexpressed in the cerebral cortex neurons of APP/PS1 mice, indicating that the overexpression of Nrf2 may be related to the antioxidant pathway of Nrf2 in neurons, but it has nothing to do with microglia. The overexpression of Nrf2 can be down-regulated by EK100, antrodin C, and ibuprofen, indicating that the overexpression of Nrf2 may be a feedback effect to protect neurons after the presence of Aβ. The treatment of EK100, antrodin C, and ibuprofen may overcome the toxicity mediated by Aβ. Therefore, the feedback effect of Nrf2 can be avoided.

The Nrf2/HO-1/NQO-1 pathway in microglia can regulate the inflammatory function of microglia and inhibit Aβ accumulation through phagocytosis [46]. The oxidation and anti-oxidation mechanisms are usually balanced by certain known elements, such as Nrf2 and HO-1. The overproduction of reactive oxygen species (ROS) and/or inhibition of antioxidant defense mechanisms may become harmful, which is called oxidative stress [47]. Previous studies reported that despite the presence of oxidative stress, the expression of nuclear Nrf2 in the brains of human AD patients is reduced [48]. However, other studies have shown that the expression of Nrf2 target genes in AD brain is increased. Previous studies have shown that Nrf2 transcripts are significantly reduced in 3-month-old APP/PS1 mice but not found in 6-month-old mice [49]. Fragoulis et al. demonstrated that in the AD mouse model, the administration of methysticin activates the Nrf2 pathway and reduces neuroinflammation, hippocampal oxidative damage, and memory loss [50]. Tanji et al. demonstrated that the expression level of HO-1 in the temporal cortex of AD patients is increased compared with the control group [51]. Other studies have reported increased NQO1 activity and immunoreactivity in the brains of AD patients associated with AD pathology [52]. A reasonable explanation for the differences in these reports is that Nrf2 levels may change during disease progression based on the degree of ROS production. It is now accepted that Nrf2 is up-regulated in the early stages of AD by Aβ-induced ROS but starts to decrease as the disease progresses [53]. Kanninen et al. used APP/PS1 mice to prove that Nrf2 expression decreases in the later stage [54]. Although the underlying mechanism has not been elucidated, the damage of the Nrf2 pathway may be related to the progression of the disease. Yammzaki et al. found that the expression of Nrf2 increased in APP/PS1 mice [55]. This may be mediated by the Aβ-mediated phosphorylation of P62, which interacts with Kelch-like ECH-associated protein 1 (Keap1). In addition, cell type-specific Nrf2 expression should be checked [56].

Aβ clearance can be mediated by microglia. In addition, Aβ can also be cleared through vascular access [57], the glymphatic system [58], and IPAD [59]. IPAD failure can lead to cerebral amyloid angiopathy (CAA), where Aβ mainly accumulates in capillaries and arterial smooth muscle cells BM [4]. Therefore, the perivascular deposition of Aβ was determined. The results showed that EK100 and antrodin C significantly increased the perivascular deposition of Aβ, indicating that EK100 and antrodin C reduced plaque deposition by promoting the clearance of Aβ through the perivascular pathway. However, partial failure of these clearance pathways can increase perivascular deposits.

It has been found that antrodin C has anti-inflammatory effects in RAW264.7 macrophages activated by LPS [20] and can effectively interfere with hyperglycemia-induced senescence and apoptosis by activating the HO-1/NQO-1-dependent cellular antioxidant defense system [21]. In our current study, the promotion of HO-1/NQO-1-dependent cellular antioxidant defense was detected in the treatment of antrodin C and ergosterol, but it was not detected during the treatment of EK-100 and ibuprofen. This may help antrodin C to reduce Aβ burden more than EK100.

EK100 also has anti-inflammatory effects [17] and has been studied to improve ischemic stroke [18]. In our current study, EK100 did not show cellular antioxidant defense that depends on HO-1/NQO-1. Indomethacin was used as a positive control to evaluate the analgesic activity of EK100 in vitro [60]. Our current study uses ibuprofen as an NSAID control and found that EK100 has a similar effect on NCA activation as ibuprofen, which means that the NSAID activity of ibuprofen and EK100 can increase the activation of NCA. In previous studies, it has been found that EK100 and its isomer ergosterol can inhibit LPS-mediated macrophage activation [17]. In our current study, we also found that both of these compounds can inhibit LPS-mediated BV2 microglia activation. However, EK100 (instead of ergosterol) inhibits the activation of non-clustered and clustered microglia, indicating that Aβ-mediated activation of microglia is specifically inhibited by EK100 but not affected by ergosterol.

It is speculated that both soluble Aβ and Aβ plaques can cause neuroinflammation and subsequent damage to the hippocampal nerves, which then leads to behavioral defects. EK100 can effectively remove plaque, reduce neuroinflammation, increase neurogenesis, and improve behavioral defects. According to this interference, antrodin C may also have the ability to promote neurogenesis.

In this study, we revealed four compounds with anti-inflammatory activity, two of which are structurally similar and exhibit different effects on AD-related pathological changes (including brain Aβ clearance). However, we cannot determine the precise target of these effects by using APP/PS1 mice as an animal model. In addition, the structure-activity relationship between EK100 and ergosterol and between antrodin C and ibuprofen has not been resolved. The ultimate goal of this study is to reveal the Aβ clearance promoting activity and the structure-activity relationship of these compounds. The biggest challenge in the future is that we need more technologies and research models, including cell and animal models, to achieve this goal.

## 4. Materials and Methods

### 4.1. Reagents 

5-Bromo-2′-deoxyuridine (BrdU), formic acid, ergosterol, and ibuprofen were purchased from Sigma-Aldrich (St Louis, MO, USA). General chemicals were purchased from Sigma-Aldrich (St Louis, MO, USA) or Merck (Darmstadt, Germany).

### 4.2. Extraction, Isolation, Purification, and Structure Determination of Compounds in A. cinnamomea

Freeze-dried powder of *A. cinnamomea* of the submerged whole broth (Batch No. MZ-247) was provided by the Biotechnology Center of Grape King Inc., Chung-Li City, Taiwan, Republic of China. The purification procedure of EK100 from *A. cinnamomea* was described previously [61]. In brief, freeze-dried powder of *A. cinnamomea* of the submerged whole broth was extracted three times with methanol at room temperature. The methanol extract was evaporated in vacuum to give a brown residue, which was suspended in H_2_O and then partitioned with 1 L of ethyl acetate. The ethyl acetate fraction was chromatographed on silica gel using mixtures of hexane and ethyl acetate of increasing polarity as eluents and further purified with high-performance liquid chromatography (HPLC). Twelve components were identified. EK100 was eluted with 10% ethyl acetate in hexane. The structure of EK100 was elucidated by mass and nuclear magnetic resonance (NMR) spectral data.

The purification procedure of antrodin C from *A. cinnamomea* mycelia was described previously [17]. In brief, the dried and ground mycelium of *A. cinnamomea* was extracted with 95% ethanol. Then, the 95% ethanol extract was concentrated under reduced pressure. The residues are suspended in H_2_O and partitioned with *n*-hexane and then ethyl acetate. The ethyl acetate fraction was sequentially chromatographed on silica gel and Sephadex LH-20 columns to obtain antrodin C. The structure of antrodin C was elucidated by mass and NMR spectral data. 

The purification procedure of anticin K, dehydroeburicoic acid, eburicoic acid, sulfurenic acid, and dehydrosulfurenic acid from *A. cinnamomea* mycelia was described previously [62]. In brief, dried *A. cinnamomea* fruiting bodies were extracted five times with methanol at 50 °C for 12 h and then concentrated under reduced pressure. The methanol extract was separated by silica gel flash column (70–230 mesh, 15 × 10 cm) with a gradient solvent system of CH_2_Cl_2_ 100% to methanol 100%, to provide 12 fractions (ACFB.1–ACFB.12). ACFB.3 was purified by preparative HPLC (Cosmosil 5C18-AR-II, 5 μm, 250 × 20 mm i.d., acetonitrile/H_2_O containing 0.1% formic acid, 35:65, flow rate 10 mL/min) to produce antcin K. ACFB.5 was purified by preparative HPLC (Cosmosil 5C_18_-AR-II, 5 μm, 250 × 20 mm i.d., acetonitrile/H_2_O containing 0.1% formic acid, 65:35, flow rate 10 mL/min) to produce four subfractions (ACFB.5.1–ACFB.5.4). ACFB.5.2 was further purified by recycled preparative HPLC (Cosmosil 5C18-AR-II, 5 μm, 250 × 20 mm i.d., acetonitrile/H_2_O containing 0.1% formic acid, 55:45, flow rate 10 mL/min, recycled 8 times) to produce dehydrosulfurenic acid. ACFB.6 was purified by preparative HPLC (Cosmosil 5C18-AR-II, 5 μm, 250 × 20 mm i.d., acetonitrile/H_2_O containing 0.1% formic acid, 70:30, flow rate 10 mL/min) to produce two subfractions: ACFB.6.1 and ACFB.6.2. ACFB.9 was purified by preparative HPLC (Cosmosil 5C18-AR-II, 5 μm, 250 × 20 mm i.d., acetonitrile /H_2_O containing 0.1% formic acid, 90:10, flow rate 10 mL/min) to produce dehydroeburicoic acid and eburicoic acid. The structure of the compounds used in this study is shown in Figure 1.

### 4.3. Cell Culture 

BV2 cells, a cell line derived from primary mouse microglia cells, were maintained in Dulbecco’s Modified Eagle’s Medium (DMEM) supplemented with 10% fetal bovine serum (FBS), 100 U/mL penicillin, and 100 μg/mL streptomycin. For treatment, the cell medium was replaced with DMEM containing 1% FBS. 

### 4.4. Measurement of Nitric Oxide

For the assessment of the amount of nitric oxide production, the Griess reagent (0.05% *N*-(1-naphthyl)-ethylene-diamine dihydrochloride, 0.5% sulfanilamide, and 1.25% phosphoric acid) was employed. The accumulated nitrite, a stable breakdown product of nitric oxide, can be recorded. The optical density was detected at a wavelength of 540 nm using a microplate reader with NaNO_2_ as standard.

### 4.5. Cell Viability Assay 

The reduction of 3-[4,5-dimethylthiazol-2-yl]-2,5-diphenyl-tetrazolium bromide (MTT) was used to evaluate cell viability. Cells were incubated with 0.5 mg⋅mL^−1^ MTT for 1 h. The formazan particles were dissolved with DMSO. OD_600nm_ was measured using an ELISA reader.

### 4.6. Animal Management and Administration

The Institutional Animal Care and Use Committee (IACUC) at the National Research Institution of Chinese Medicine approved the animal protocol (IACUC No: 106-417-4). All experimental procedures involving animals and their care were carried out in accordance with The Guide for the Care and Use of Laboratory Animals published by the United States National Institutes of Health. APP/PS1 was purchased from Jackson Laboratory (No. 005864). The breeding gender ratio was a male with two females in one cage. Experiments were conducted using wild-type siblings and APP/PS1 transgenic C57BL/6J mice. The animals were housed under controlled room temperature (24 ± 1 °C) and humidity (55–65%) with a 12:12 h (07:00–19:00) light–dark cycle. All mice were provided with commercially available rodent normal chow diet and water *ad libitum*. 

The dose of EK100 used for humans is typically 7380 mg per day. The mouse dose used was converted from a human-equivalent dose (HED) based on body surface area according to the US Food and Drug Administration formula. Assuming a human weight of 60 kg, the HED for 7380 (mg)/60 (kg) = 123 × 12.3 = 10 mg/kg; the conversion coefficient of 12.3 is used to account for differences in body surface area between mice and human, as previously described [63]. To investigate the effect of EK100 in CNS, 3 doses (30 mg kg^−1^ day^−1^) were applied. EK100, ergosterol, antrodin C, and ibuprofen were dissolved in vehicle (10% Kolliphor EL, 5% ethanol, 85% dextrose solution (5% in water), pH 7.2) to get a final concentration of 3 mg/mL. For studying the therapeutic effect, thirty APP/PS1 mice (5 months of age) were randomly assigned to five groups (*n* = 6, half male and half female) and were administrated by gavage with vehicle, EK100, antrodin C, ergosterol, and ibuprofen (30 mg⋅kg^−1^⋅day^−1^) for one month. Ergosterol was used as EK100’s structural control and ibuprofen was used as an NSAID control. Nesting task was performed 30 days after drug administration. The animals were sacrificed on the 31st day after drug administration, and then, the Aβ-related pathological changes were inspected by immunohistochemistry and immunoblotting. For neurogenesis measurement, BrdU was injected intraperitoneally at 50 mg⋅kg^−1^⋅day^−1^ during the last 7 days.

### 4.7. Nesting Test 

After oral gavage administration for 30 days, mice were assessed for a nesting test as described previously. In brief, two Nestlets (5 g) were placed into cage at 1 h before the dark cycle, and then, the nest score and the weight of unshredded Nestlets were determined after overnight. Nest construction was scored using a six-graded scale [64]. A score of 0 indicates undisturbed Nestlet; 1, Nestlet was disturbed, but nesting material has not been gathered to a nest site in the cage; 2, a flat nest; 3, a cup nest; 4, an incomplete dome and 5, a complete and enclosed dome.

### 4.8. Tissue Processing 

Mice after being anesthetized were sacrificed by transcardial saline perfusion. The mouse brain was removed, and half of the brain was homogenized in homogenization buffer containing 20 mM Tris-HCl (pH 7.4), 320 mM sucrose, 2 mM ethylene diamine tetraacetic acid, 1 mM phenylmethylsulfonyl fluorid, 5 μg·mL^−1^ leupeptin, and 5 μg·mL^−1^ aprotinin. Another half brain was immersed in 4% formaldehyde overnight at 4 °C and cryoprotected. Then, brain tissue was sectioned into 30 μm thick sections. Three slides spanning approximately bregma-1.58 to -1.82 in each brain were used for staining and analysis.

### 4.9. Amylo-Glo Staining 

Staining for fibrillary amyloid was performed using Amylo-Glo as described by the manufacturer (Biosensis Inc., Thebarton, South Australia).

### 4.10. Immunohistochemistry 

Immunohistochemistry was performed as described previously [29]. Briefly, sections were blocked in phosphate buffer saline (PBS) containing 1% bovine serum albumin, 3% normal donkey serum, and 0.3% Triton X-100 for 1 h. Then, they were incubated in PBS containing 1% bovine serum albumin, 1% normal donkey serum, 0.3% Triton X-100, and primary antibodies, including mouse monoclonal antibodies to Aβ1-16 (AB10, Millipore, MAB5208), glial fibrillary acidic protein (GFAP, Millipore, MAB5804), heme oxygenase 1 (HO-1, Santa Cruz, sc-390991), NAD(P)H quinone dehydrogenase 1 (NQO-1, Santa Cruz, sc-376023), BrdU (Santa Cruz, sc-32323), goat polyclonal antibody to anti-ionized calcium-binding adaptor molecule-1 (Iba-1) antibody (abcam, ab5076), and rabbit polyclonal antibody to anti-Nrf2 antibody (abcam, ab137550), and doublecortin (DCX, abcam, ab18723) overnight at 4 °C. Then, sections were incubated in antibody dilution buffer containing Hoechst33258 (Invitrogen, 2 μg mL^−1^), fluorescein isothiocyanate, rhodamine red X-conjugated donkey anti-mouse IgG, rhodamine red X-conjugated donkey anti-rabbit IgG, or Alexa Fluor 647-conjugated donkey anti-goat IgG (Jackson ImmunoResearch, 705-605-147) at room temperature for 2 h. After being washed with PBS containing 0.01% Triton X-100, sections were mounted with Aqua Poly/Mount (Polyscience Inc., Warrington, PA, USA) for microscopic analysis using a Zeiss LSM 780 confocal microscopy (Jena, Germany). Representative confocal images had 10 μm depth with maximal projection. Quantification of amyloid plaque was performed using ImageJ (http://imagej.nih.gov/ij/, accessed on 1 September 2021) software.

Amyloid plaque, astrocytes, and microglia were co-immunostained using anti-Aβ1-16 antibody (AB10), anti-GFAP antibody, and anti-Iba-1 antibody, respectively. On amyloid plaque staining, we observed a wide range of plaque size distribution from 2 to 219 pixels using MetaMorph software. To exclude the non-specific staining and the oversize plaque, the plaques with a size range of 11 to 80 pixels were included for observation. For determining perivascular Aβ deposition, after merging the images of the GFAP and Aβ channels into one, the perivascular Aβ deposition was determined by calculating the positive pixels present in the GFAP vascular mask in the image. 

Activated glia is a prominent feature of AD neuropathology, with both reactive astrocytes and activated microglia clustering around amyloid plaques [41]. Therefore, the number of clusters with plaque-associated astrocytes (PAA) and plaque-associated microglia (PAM) were determined by staining with GFAP-antibody and Iba-1-antibody, respectively. The reduced number of clusters with PAM may be related to the reduced number of plaques. To verify the relationship of cluster size with plaque size, the size alteration of the individual plaque-associated microglia cluster was analyzed by scatter plot. 

### 4.11. Skelecton Analysis 

The number of branches, junctions, and endpoints of microglia were quantified using FIJI ImageJ software [32]. Briefly, the images of brain sections were transformed to 8-bit format, and the Unsharp Mask and Despeckle were applied to increase contrast and remove noise; then, the images were skeletonized and analyzed using the plug-in AnalyzeSkeleton (2D/3D). 

### 4.12. Quantification of AB10-Stained Plaques

The quantification of AB10-stained plaques was conducted. At least 3 coronal brain sections from each mouse were used for analysis. Each image was adjusted to the threshold for pixel detection (threshold setting for AB10-positive signal is 200). To eliminate background, particle large than 80 pixels (approximately 56 μm^2^) less than 20 pixels (approximately 14 μm^2^) was excluded. 

### 4.13. Measurement of Aβ Levels

Two-step sequential extraction of the brain Aβ using 2% sodium dodecyl sulfate (SDS) and 70% formic acid (FA; Sigma) was processed as described previously [30]. Briefly, cortical homogenate was mixed with an equal volume of 4% SDS in homogenization buffer containing protease inhibitor. Then, the sample was sonicated and centrifuged at 100,000× *g* for 60 min at 4 °C. The supernatant was considered an SDS-soluble fraction. The SDS-insoluble pellet was further suspended in 70% FA and centrifuged at 100,000× *g* for 60 min at 4 °C. The supernatant was collected and neutralized with 1 M Tris, pH 11. SDS-soluble and SDS-insoluble fractions were stored at −80 °C until sandwich enzyme-linked immunosorbent assay (ELISA) analysis. Aβ level was measured by a sensitive sandwich ELISA assay using a kit (Invitrogen KHB3481 and KHB3442). The detailed experiments were performed according to the manufacturer’s protocol.

### 4.14. Statistical Analysis 

Results are expressed as mean ± standard error of the mean (S.E.M) and processed for statisyical analysis using GraphPad Prism 6 software (La Jolla, CA92037 USA). The parametric data were analyzed by unpaired two-tailed Student’s t test or one way analysis of variance (ANOVA) with post hoc multiple comparisons with a Bonferroni test.

## 5. Conclusions

Our in vitro experiments have shown that EK100, antrodin C, and ergosterol have anti-inflammatory activity by detecting the production of nitric oxide. Ibuprofen is an NSAID that can specifically block the COX-2 enzyme. On the other hand, our in vivo experiments show that EK100 and antrodin C reduce the formation of amyloid plaques in the cortex and hippocampus, which may be due to the glial phagocytosis and perivascular granule pathways in APP/PS1 mice to promote Aβ clearance. We also showed that EK100 and antrodin C ameliorated behavioral deficits in APP/PS1 mice. EK100 also promotes hippocampal neurogenesis. These findings raise the possibility that EK100 and antrodin C may have the potential to treat AD.

## Figures and Tables

**Figure 1 ijms-22-10413-f001:**
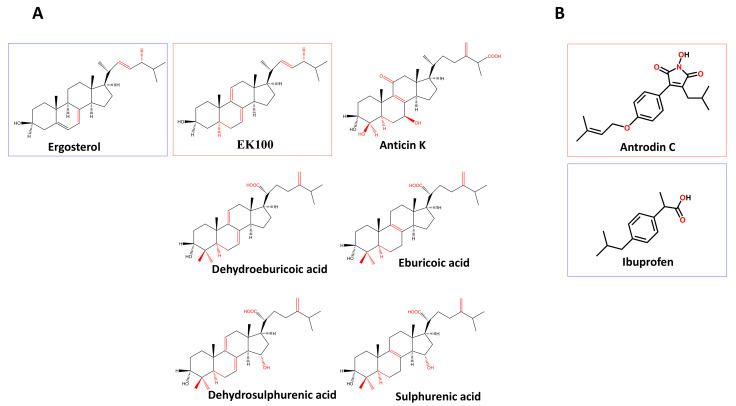
The structure of the compounds used in this study. (**A**) The structure of EK100 (Ergosta-7,9(11),22-trien-3β-ol), ergosterol (Ergosta-5,7,22-trien-3β-ol), anticin K (3α,4β,7β-trihygroxy-4α-methylergosta-8,24(28)-dien-11-on-26-oic acid), dehydroeburicoic acid ((2*R*)-2-[(3*S*,5*R*,10*S*,13*R*,14*R*,17*R*)-3-hydroxy-4,4,10,13,14-pentamethyl-2,3,5,6,12,15,16,17-octahydro-1*H*-cyclopenta[a]phenanthren-17-yl]-6-methyl-5-methylideneheptanoic acid), eburicoic acid ((2R)-2-[(3S,5R,10S,13R,14R,17R)-3-hydroxy-4,4,10,13,14-pentamethyl-2,3,5,6,7,11,12,15,16,17-decahydro-1H-cyclopenta[a]phenanthren-17-yl]-6-methyl-5-methylideneheptanoic acid), dehydrosulfurenic acid ((2*R*)-2-[(3*S*,5*R*,10*S*,13*R*,14*R*,15*S*,17*R*)-3,15-dihydroxy-4,4,10, 13,14-pentamethyl-2,3,5,6,11,12,15,16,17-decahydro-1*H*-cyclopenta[a] phenanthren-17-yl]-6-methyl-5-methylideneheptanoic acid), and sulfurenic acid ((2*R*)-2-[(3*S*,5*R*,10*S*,13*R*,14*R*,15*S*,17*R*)-3,15-dihydroxy-4,4,10,13,14-pentamethyl-2,3,5,6,7, 11,12,15,16,17-decahydro-1*H*-cyclopenta[a]phenanthren-17-yl]-6-methyl-5-methyl-ideneheptanoic acid). (**B**) The structure of antrodin C (3-isobutyl-4-[4-(3-methyl-2-butenyloxy)phenyl]-1*H*-pyrrol-1-ol-2,5-dione) and ibuprofen (isobutylphenyl-propionic acid). The red squares indicate the bioactive compounds and the blue squares indicate the control compounds.

**Figure 2 ijms-22-10413-f002:**
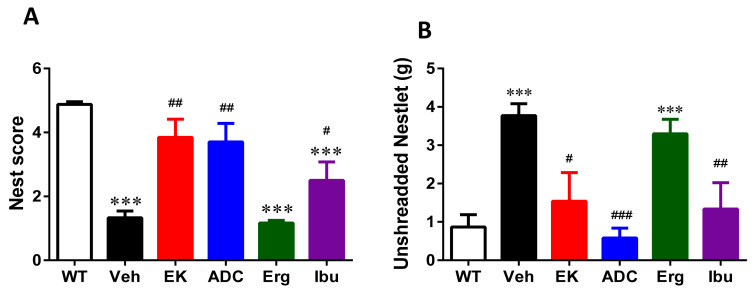
EK100, antrodin C, and ibuprofen ameliorate nesting behavior deficits in APP/PS1 mice. APP/PS1 transgenic mice were orally administered with vehicle (Veh) or EK100 (EK), antrodin C (ADC), ergosterol (Erg), and ibuprofen (Ibu) (30 mg⋅kg^−1^⋅day^−1^, *n* = 6 each). Nesting tasks were performed at 30 days post administration. The image in wild-type (WT) mice (*n* = 6) is also compared. Bar graphs show the results from the nesting task’s nest score (**A**) and unshredded Nestlet (**B**) from nesting task. The results are the mean ± S.E.M. Significant differences between WT group and the other groups are indicated by ***, *p* < 0.001. Significant differences between Veh group and reagent-treated groups are indicated by ^#^, *p* < 0.05; ^##^, *p* < 0.01; ^###^, *p* < 0.001.

**Figure 3 ijms-22-10413-f003:**
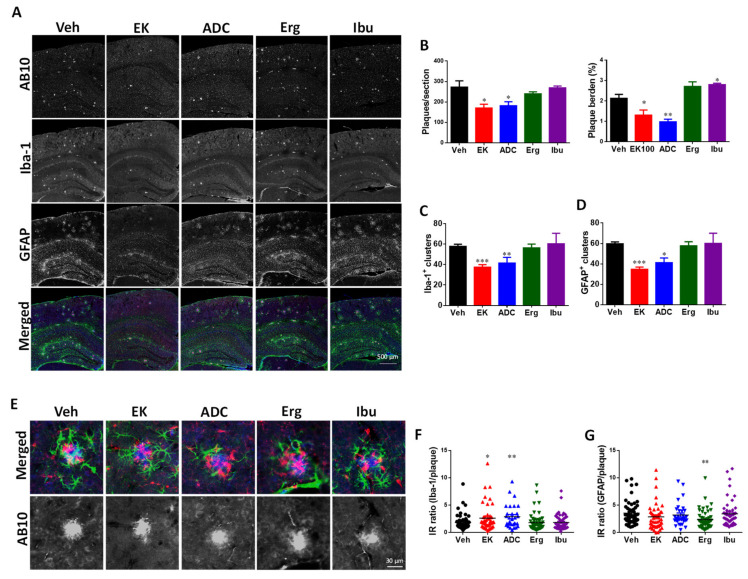
EK100 and antrodin C reduce amyloid plaque load, number of glial clusters, and plaque-related glial activation in APP/PS1 mice. APP/PS1 transgenic mice orally administered vehicle (Veh) or EK100 (EK), antrodin C (ADC), ergosterol (Erg), and ibuprofen (Ibu) (30 mg⋅kg^−1^⋅day^−1^, each *n* = 6) 1 months, and then amyloid plaques, microglia, and astrocytes were immunostained with AB10, Iba-1, and GFAP antibodies, respectively. (**A**). The representative fluorescent images of AB10 (blue in the merged panel), Iba-1 (red in the merged panel), and GFAP (green in the merged panel) are shown. Scale bar: 500 μm. (**B**–**D**), The number and burden of AB10-stained plaque (**B**) and the number of Iba-1^+^ clusters (**C**) and GFAP^+^ clusters (**D**) in cerebral hemisphere are counted and shown. The results are the mean ± S.E.M. Significant differences between the Veh group and reagent-treated groups are indicated by *, *p* < 0.05; **, *p* < 0.01; ***, *p* < 0.001. (**E**). The representative fluorescent images of glial clusters immunostained with AB10 (blue in the merged panel), Iba-1 (red in the merged panel), and GFAP (green in the merged panel) antibodies are shown. Scale bar: 30 μm. (**F**,**G**). The immunointensity (IR) ratio of Iba-1 to AB10 (**F**) and GFAP to AB10 (**G**) are calculated and shown. The results are the mean ± S.E.M. Significant differences between Veh group and reagent-treated groups are indicated by *, *p* < 0.05; **, *p* < 0.01.

**Figure 4 ijms-22-10413-f004:**
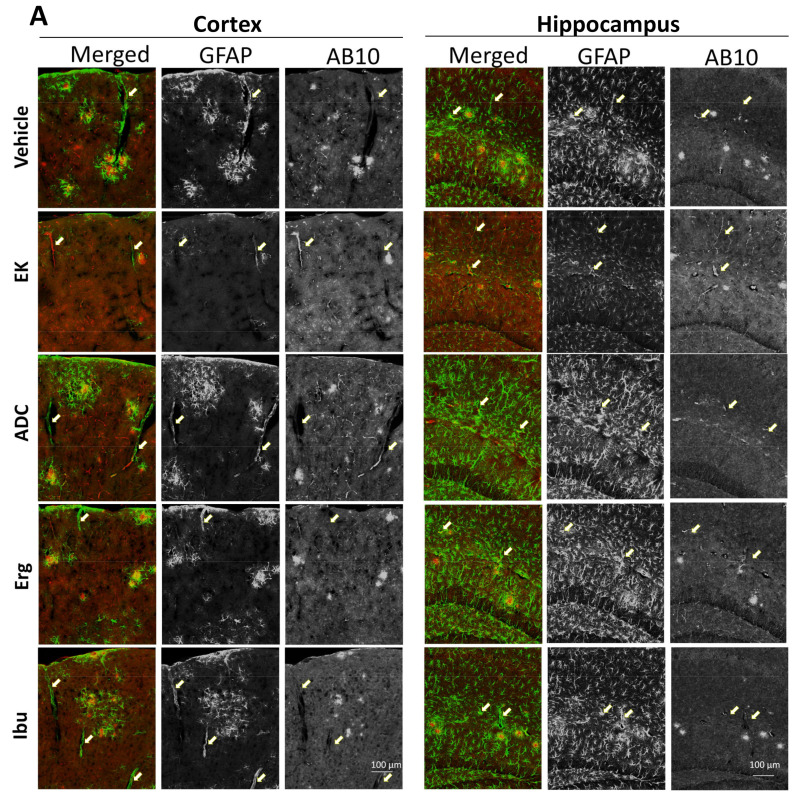
EK100 and antrodin C increase perivascular Aβ deposition in APP/PS1 mice. APP/PS1 transgenic mice orally administered vehicle (Veh) or EK100 (EK), antrodin C (ADC), ergosterol (Erg), and ibuprofen (Ibu) (30 mg⋅kg^−1^⋅day^−1^, each *n* = 6) for 1 months, and then astrocytes and microglia were immunostained GFAP and Iba-1 antibodies, respectively. (**A**) The representative immunostaining images of GFAP and AB10 in the cortex and hippocampus are shown. Scale bar: 100 μm. Arrows indicate the representative perivascular area. (**B**). The percentage of perivascular Aβ deposition to total Aβ deposition is calculated by MetaMorph image analysis software and shown. The results are the mean ± S.E.M. Significant differences between the Veh group and the other groups are indicated by *, *p* < 0.05; **, *p* < 0.01.

**Figure 5 ijms-22-10413-f005:**
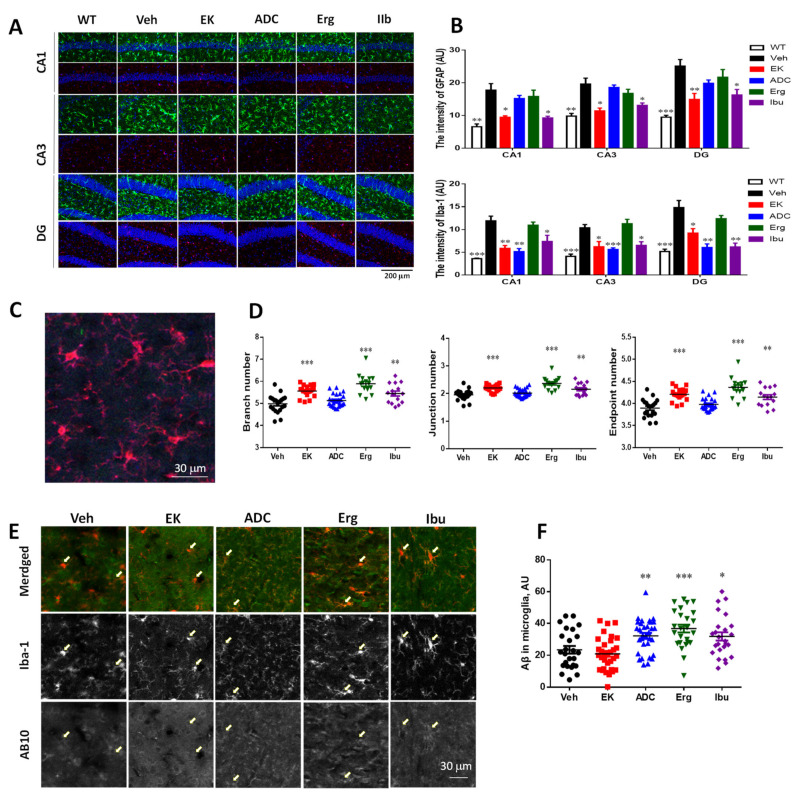
EK100 reduce non-clustered glial activation in APP/PS1 mice. APP/PS1 transgenic mice orally administered vehicle (Veh) or EK100 (EK), antrodin C (ADC), ergosterol (Erg), and ibuprofen (Ibu) (30 mg⋅kg^−1^⋅day^−1^, each *n* = 6) for 1 month, and then astrocytes and microglia were immunostained with GFAP and Iba-1 antibodies, respectively. (**A**,**B**) The representative immunostaining images of GFAP (green) and Iba-1 (red) without being associated with plaque in Cornu Amonis (CA)1, CA3, dentate gurus (DG) are shown in (**A**). The image in wild-type (WT) mice (*n* = 6) is also compared. Scale bar: 200 μm. The level of GFAP and Iba-1 in the CA1, CA3, and DG are calculated by MetaMorph image analysis software and shown in (**B**). The results are the mean ± S.E.M. Significant differences between the Veh group and the other groups are indicated by *, *p* < 0.05; **, *p* < 0.01; ***, *p* < 0.001. (**C**,**D**). The representative image of Iba-1 (red) not associated with plaque is shown in (**C**). The branch number, junction number, and end point number of Iba-1 immunostained microglia are calculated by FIJI ImageJ software. The results are the mean ± S.E.M. Significant differences between the Veh group and the other groups are indicated by **, *p* < 0.01; ***, *p* < 0.001. (**E**,**F**). The representative immunostaining images of Iba-1 (red) and Aβ not associated with plaque in the cortex are shown in (**E**). Arrows indicate the representative microglia. The intracellular Aβ in microglia are calculated by MetaMorph image analysis software and shown in (**F**). The results are the mean ± S.E.M. Significant differences between the Veh group and the other groups are indicated by *, *p* < 0.05; **, *p* < 0.01; ***, *p* < 0.001.

**Figure 6 ijms-22-10413-f006:**
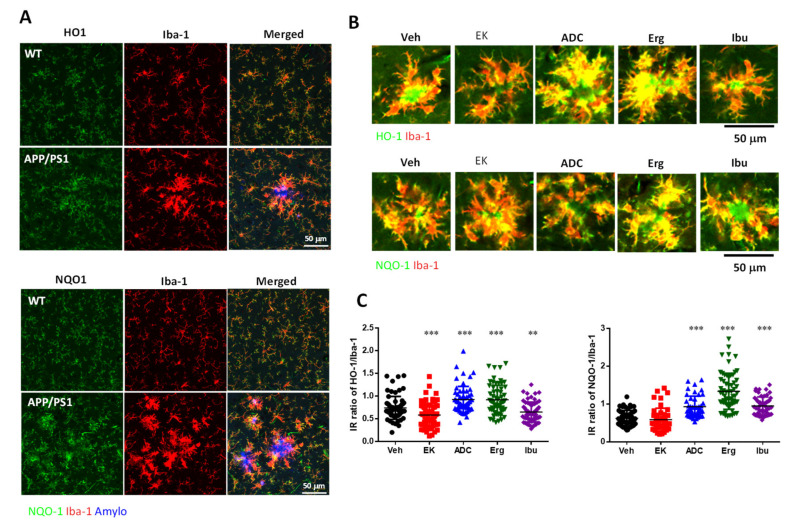
Antrodin C, but not EK100, promotes HO-1 in APP/PS1 mice. (**A**–**E**). APP/PS1 transgenic mice orally administered vehicle (Veh) or EK100 (EK), antrodin C (ADC), ergosterol (Erg), and ibuprofen (Ibu) (30 mg⋅kg^−1^⋅day^−1^, each *n* = 6) for 1 month. The representative images of HO-1 (green in upper panel) and NQO1 (green in lower panel) in Iba-1^+^ microglia (red) of WT mice and APP/PS1 transgenic mice are shown in (**A**). Amyloid plaques were stained with Amylo-Glo (blue). Scale bar: 50 μm. The representative images of HO-1 (green in upper panel) and NQO1 (green in lower panel) in Iba-1^+^ microglia (red) of APP/PS1 transgenic mice are shown in (**B**). Immunointensity (IR) ratio of HO-1/Iba-1 and NQO-1/Iba-1 are calculated and shown in (**C**). Representative immunoblots of Nrf2 and HO-1 and β-actin of cortical homogenate are shown in (**D**). The protein level in wild-type (WT) mice (*n* = 6) is also compared. The ratio of Nrf2 and HO-1 to β-actin is presented as a percentage of the WT group (**E**). (**F**,**G**) APP/PS1 transgenic mice orally administered vehicle (Veh) or EK100 (EK) and antrodin C (ADC) (30 mg⋅kg^−1^⋅day^−1^, each *n* = 6) for 1 months. The representative images of Nrf2 (green in merged panel) and AB10 (red in merged panel) in neurons of WT mice and APP/PS1 transgenic mice are shown in (**F**). Nuclei staining using Hoechst 33258 (blue). Scale bar: 20 μm. Immunointensity (IR) of Nrf2 and AB10 were calculated and shown in (**G**). The results are the mean ± S.E.M. Significant differences between the Veh group and the other groups are indicated by *, *p* < 0.05; **, *p* < 0.01; ***, *p* < 0.001.

**Figure 7 ijms-22-10413-f007:**
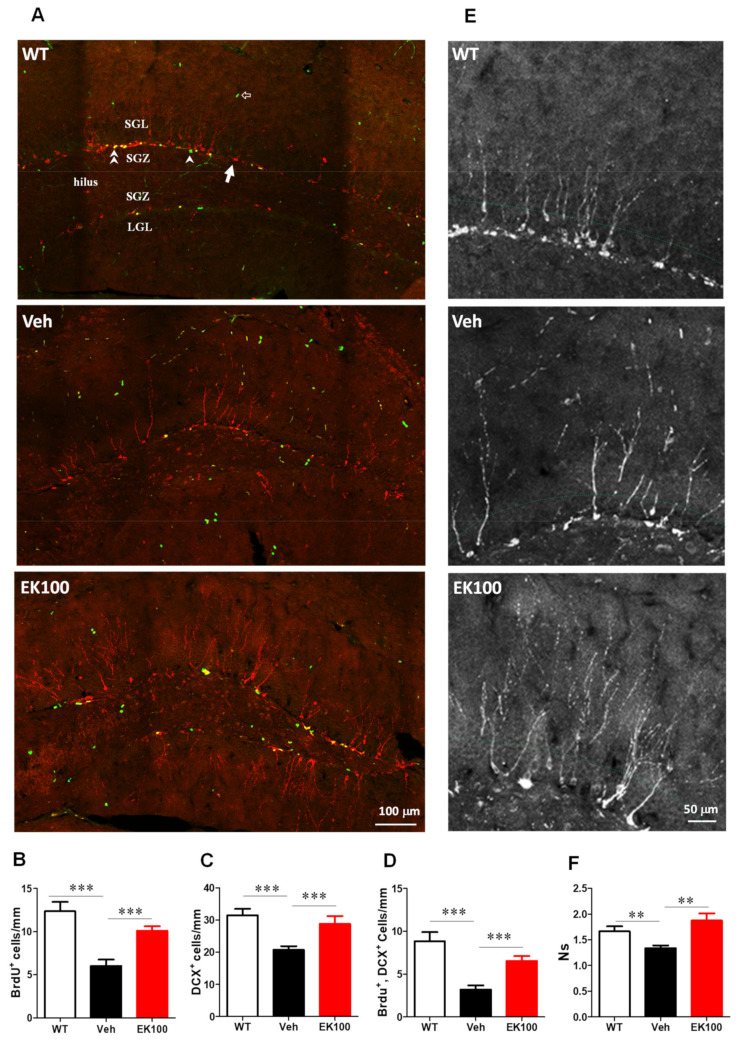
EK100 promotes hippocampal neurogenesis in APP/PS1 mice. APP/PS1 transgenic mice orally administered with vehicle (Veh) or EK100 (30 mg⋅kg^−1^⋅day^−1^, *n* = 6 each) for 1 month. Wild-type (WT) mice were used as non-transgenic control. Hippocampal neurogenesis was detected by immunohistochemical staining with doublecortin antibody (DCX, red) and BrdU antibody (green). The representative images of the dentate gyrus are shown in (**A**). Arrow indicated DCX-labeled newly born neuron; arrow head indicates proliferating type 2 neuroprogenitor; double arrow head indicates the newly born neuron immediately after proliferation; hollow arrow indicates proliferating cells other than the neuroprogenitor. ML, molecular layer; UGL, upper blade granular cell layer; SGZ, subgranular zone; LGL, lower blade granular cell layer. Scale bar: 100 μm. Panels **B**–**D** show the number/mm SGZ of BrdU positive cells (BrdU^+^, **B**); DCX-positive cell (DCX^+^, **C**) and the cells with double labeling (BrdU^+^, DCX^+^, **D**). (**E**) The representative immunostaining images of the upper blade dentate gyrus area are shown. Scale bar: 50 μm. Secondary dendrites of the DCX^+^ cells are counted along the middle of granular cell layer (GCL, green dashed line). (**F**) The dendritic complexity was analyzed by laminar quantification. The branch ratio of secondary dendrites of DCX^+^ cells were shown. Dendrites of DCX^+^ cells are counted along the middle of the GCL. The results are the mean ± S.E.M. Significant differences between the Veh group and the other groups. **, *p* < 0.01; ***, *p* < 0.001.

**Table 1 ijms-22-10413-t001:** Anti-inflammatory activity of four reagents on LPS-activated BV2 cells.

	Nitric Oxide	MTT Reduction
Vehicle	25.99 ± 1.14	97.62 ± 1.41
CA-Et		
50 μg/mL	10.55 ± 1.60 ***	101.10 ± 3.39
100 μg/mL	4.77 ± 0.68 ***	100.20 ± 1.54
Anticin K		
50 μM	21.01 ± 0.96	83.36 ± 10.19 **
100 μM	21.16 ± 1.14	89.58 ± 5.50
EA/DEA		
10 μg/mL	24.22 ± 3.17	45.37 ± 3.07 ***
20 μg/mL	16.20 ± 0.49 **	31.94 ± 2.585 ***
SA/DSA		
10 μg/mL	26.26 ± 1.35	73.70 ± 4.31 ***
20 μg/mL	17.46 ± 2.87 *	55.16 ± 2.07 ***
EK100		
10 μM	16.84 ± 1.38 ***	100.9 ± 2.07
20 μM	17.19 ± 1.24 **	96.19 ± 3.14
Antrodin C		
50 μM	18.54 ± 1.83 ***	102.7 ± 3.42
100 μM	15.61 ± 1.01 ***	104.8 ± 2.82
Ergosterol		
5 μM	16.29 ± 1.64 ***	89.14 ± 2.59
10 μM	15.55 ± 1.38 ***	82.30 ± 0.91 **
Ibuprofen		
50 μM	20.27 ± 0.99	103.3 ± 3.56
100 μM	21.80 ± 0.59	101.9 ± 0.63

CA-Et, ethanol extract of *A. cinnamomea* mycelium; EA/DEA, the mixture of eburic acid and dehydroeburic acid; SA/DSA, the mixture of sulfurenic acid and dehydrosulfurenic acid. Significant differences between Vehicle group and reagent-treated groups are indicated by *, *p* < 0.05; **, *p* < 0.01; ***, *p* < 0.001.

**Table 2 ijms-22-10413-t002:** The effects of four compounds in APP/PS1 mice.

	EK100	ADC	Erg	Ibu	WT
Pathological improvement					
1. Nesting behavior:					
Nesting score	↑	↑	–	↑	↑
Unthreaded nestlet	↓	↓	–	↓	↓
2. Aβ deposition:					
Plaque number	↓	↓	–	–	ND
Plaque burden	↓	↓	–	↑	ND
Putative Aβ clearance pathways and neuroprotective effects					
1. Perivascular pathway					
Perivascular Aβ	↑	↑	–	–	ND
2. PAM/PAA pathway					
Iba-1^+^ cluster number	↓	↓	–	–	ND
Iba-1 IR in cluster	↑	↑	–	–	ND
GFAP^+^ cluster number	↓	↓	–	–	ND
GFAP IR in cluster	–	–	↓	–	ND
HO-1/Iba-1 IR ratio	↓	↑	↑	↓	ND
NQO-1/Iba-1 IR ratio	–	↑	↑	↑	ND
3. NCM/NCA pathway					
Iba-1 IR in HNCM	↓	↓	–	↓	↓
Aβ IR in HNCM	–	↑	↑	↑	ND
Microglia ramification	↑	–	↑	↑	ND
GFAP IR in HNCA	↓	–	–	↓	↓
4. Neuron protection					
Total Nrf2	↓	↓	–	↓	↓
Neuronal Nrf2	↓	↓	ND	ND	↓
Neurogenesis	↑	ND	ND	ND	↑

ADC, antrodin C; erg, ergosterol; Ibu, Ibuprofen; WT, wild-type mice; Aβ, amyloid β; Iba-1, ionized calcium-binding adaptor molecules; GFAP, glial fibrillary acidic protein; PAM, plaque-associated microglia; PAA, plaque-associated astrocytes; HO-1, heme oxygenase-1; NQO-1, NAD(P)H quinone dehydrogenase 1; IR, immunoreactivity; NCM, non-clustered microglia; NCA, non-clustered astrocytes; Nrf2, nuclear factor erythroid-2 related factor 2; ↑, increased; ↓, decreased; –, no effect; ND, not determined.

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
