# Peer review of "EK100 and Antrodin C Improve Brain Amyloid Pathology in APP/PS1 Transgenic Mice by Promoting Microglial and Perivascular Clearance Pathways"

_ijms, 2021, doi:10.3390/ijms221910413_

Round 1

Reviewer 1 Report

The authors investigate the role of EK100 and Antrodin C on AD pathology. Using both in vitro and in vivo approaches, they confirm anti-inflammatory effects of the compounds and provide new evidence of their contribute in reducing the formation of Amyloid plaques and behavioral deficits in APP/PS1 mice. They also reported that EK100 promotes hippocampal neurogenesis. The overall findings indicate that the combined use of a EK100 and Antrodin C could be beneficial in AD treatment.

The work is interesting and well-written. I have just few modifications to suggest:

- In the introduction section, from lane 73 to 95, the description of components isolated from A. Cinnamomea is confusing. I would suggest a more organized explanation of structure and activities of these anti-inflammatory molecules. Figure 1 could be inserted in this section.

-Please specify that BV-2 cells are derived from primary mouse microglia cells in the main text and in material and method section.

-It is not clear from the text how many hours of incubation with LPS were carried out before MTT assay. If the incubation time used was shorter than 24hrs, I suggest to verify the toxicity of the compounds even after 24h exposure.

- Figure 4 show GFAP and Iba-1 immunostaining, as correctly written in the caption. Please, change AB10 with Iba-1 in cortex and hippocampus images.

- Table 2 is missing, please add it in the main text or in supplementary materials.

Author Response

Point 1: In the introduction section, from lane 73 to 95, the description of components isolated from A. Cinnamomea is confusing. I would suggest a more organized explanation of structure and activities of these anti-inflammatory molecules. Figure 1 could be inserted in this section.

Response 1: Thanks for your kind comment, I have rewritten the section to explain the structure and activities of these anti-inflammatory molecules (line 77-89). Figure 1 is inserted in this section.

Point 2: Please specify that BV-2 cells are derived from primary mouse microglia cells in the main text and in material and method section.

Response 2: Thanks for your kind comment, I have specified that BV-2 cells are derived from primary mouse microglia cells in the main text (line 124) and in material and method section (line 561).

Point 3: It is not clear from the text how many hours of incubation with LPS were carried out before MTT assay. If the incubation time used was shorter than 24hrs, I suggest to verify the toxicity of the compounds even after 24h exposure.

Response 3: Thanks for your kind comment, I have specified the incubation time for LPS is 24h (Line 149). The toxicity of compounds were also determined after 24h.

Point 4: Figure 4 show GFAP and Iba-1 immunostaining, as correctly written in the caption. Please, change AB10 with Iba-1 in cortex and hippocampus images.

Response 4: Thanks for your kind comment, I have change AB10 with Iba-1 in the caption of Figure 4 (Line 232).

Point 5: Table 2 is missing, please add it in the main text or in supplementary materials.

Response 5: Thanks for your kind comment, I have add Table 2 in the main text (Line 409-414).

Reviewer 2 Report

In this paper titled “EK100 and antrodin C improve brain amyloid pathology in APP/PS1 transgenic mice by promoting microglial and perivascular clearance pathways” Tsay et al. studied in exploring the effects of EK100 and antrodin C isolated from Antrodia cinnamonmea on the AD-like pathology of APPswe/PS1dE9 transgenic mice. They have shown the improvement in the nesting behavior of mice, reduction in the number, burden of amyloid plaques, the activation of glial cells, and promotion in the perivascular deposition of Aβ in the brain of mice. They further reported the activation of astrocytes, regulation of microglia morphology, and promotion of plaque-associated microglia to express oxidative enzymes by EK100 and antrodin C. As a whole, their data indicated that EK100 and antrodin C reduce the pathology of AD by reducing amyloid deposits and promoting nesting behavior in APPswe/PS1dE9 mice through microglia and perivascular clearance. This manuscript is well written and all results and discussions are well aligned to the conclusion.

Author Response

Point 1: In this paper titled “EK100 and antrodin C improve brain amyloid pathology in APP/PS1 transgenic mice by promoting microglial and perivascular clearance pathways” Tsay et al. studied in exploring the effects of EK100 and antrodin C isolated from Antrodia cinnamonmea on the AD-like pathology of APPswe/PS1dE9 transgenic mice. They have shown the improvement in the nesting behavior of mice, reduction in the number, burden of amyloid plaques, the activation of glial cells, and promotion in the perivascular deposition of Aβ in the brain of mice. They further reported the activation of astrocytes, regulation of microglia morphology, and promotion of plaque-associated microglia to express oxidative enzymes by EK100 and antrodin C. As a whole, their data indicated that EK100 and antrodin C reduce the pathology of AD by reducing amyloid deposits and promoting nesting behavior in APPswe/PS1dE9 mice through microglia and perivascular clearance. This manuscript is well written and all results and discussions are well aligned to the conclusion.

Response 1: Thanks for your kind comment.